# Role of Fungi in Biodegradation of Imidazolium Ionic Liquids by Activated Sewage Sludge

**DOI:** 10.3390/molecules28031268

**Published:** 2023-01-28

**Authors:** Joanna Klein, Justyna Łuczak, Anna Brillowska-Dąbrowska

**Affiliations:** 1Department of Process Engineering and Chemical Technology, Faculty of Chemistry, Gdańsk University of Technology, ul. Narutowicza 11/12, 80-233 Gdańsk, Poland; 2Department of Molecular Biotechnology and Microbiology, Faculty of Chemistry, Gdańsk University of Technology, ul. Narutowicza 11/12, 80-233 Gdańsk, Poland

**Keywords:** Ionic liquids, imidazolium, biodegradation, bioaugumentation, fungi, sewage sludge

## Abstract

Ionic liquids (ILs), due to their specific properties, can play the role of persistent water contaminants. Fungi manifest the ability to decompose hardy degradable compounds, showing potential in the biodegradation of ILs, which has been studied extensively on sewage sludge; however, attention was drawn mainly to bacterial and not fungal species. The aim of the research was to determine the significance of fungi in ILs’ biodegradation to extend the knowledge and possibly point out ways of increasing their role in this process. The research included: the isolation and genetic identification of fungal strains potentially capable of [OMIM][Cl], [BMIM][Cl], [OMIM][Tf_2_N], and [BMIM][Tf_2_N] degradation, adjustment of the ILs concentration for biodegradability test by MICs determination and choosing strains with the highest biological robustness; inoculum adaptation tests, and finally primary biodegradation by OECD 301F test. The study, conducted for 2 mM [OMIM][Cl] as a tested substance and consortium of microorganisms as inoculum, resulted in an average 64.93% biodegradation rate within a 28-day testing period. For the individual fungal strain (*Candida tropicalis*), the maximum of only 4.89% biodegradation rate was reached in 10 days, then inhibited. Insight into the role of fungi in the biodegradation of ILs was obtained, enabling the creation of a complex overview of ILs toxicity and the possibilities of its biological use. However, only an inoculum consisting of a consortium of microorganisms enriched with a selected strain of fungi was able to decompose the IL, in contrast to that consisting only of an individual fungal strain.

## 1. Introduction

Ionic liquids (ILs), often called designer solvents, became attractive in new clean technologies as they possess very low, almost negligible vapor pressure and may be designed to possess high thermal stability and good miscibility with both polar and non-polar compounds [1,2,3]. However, their low vapor pressure does not exclude the possibility of their harmful impact on the environment; for example, due to their solubility in water, they may pose a threat to water bodies. Moreover, considering the fact that ILs may be highly stable in water, they may also become persistent water pollutants. In conclusion, the potential risk involved in the possibility of ILs’ infiltration into wastewater streams or aquatic bodies should be considered [4].

In the literature concerning ILs’ biological activity, two kinds of information can be found. First referring to their toxicity towards model microorganisms and second to biodegradability. The biodegradability of those neoteric solvents was mostly evaluated with the application of standard tests according to OECD norms [4,5,6]. Docherty et al. [4] revealed that imidazolium ILs substituted with hexyl and octyl side chains were only partially decomposed, while compounds based on pyridinium cation were entirely subjected to mineralization. The research of Garcia et al. [7] showed that dialkylimidazolium ILs are poorly biodegradable; nevertheless, the presence of an ester bond within the side chain of IL significantly increased its biodegradability. Stolte et al. [8], as well as Docherty et al. [4], also observed full biodegradation of imidazolium and pyridinium cations substituted with an octyl side chain. The suggested path of decomposition involved the oxidation of the alkyl chain terminal methyl group to alcohol, followed by further oxidation to aldehydes and carboxylic acids [9].

The results of the toxicity studies conducted so far mostly depend on the chemical structure of ILs and the type of organism applied [10]. The most general conclusion of all the tests is that the toxicity of neoteric solvents increases with the increasing alkyl chain. The tests were mostly conducted with bacteria such as marine luminescent *Vibrio fischeri* [7,9,11,12,13,14], algae like *Selenastrum capricornutum* [8,15,16,17,18], invertebrates such as *Daphnia magna* [7,11,12,19,20,21,22,23,24,25,26,27] vertebrates like Zebrafish *Danio rerio* [11,28] and plants [11,13,29,30]. Nevertheless, it has to be emphasized that all of the mentioned tests were performed applying pure cultures of model microorganisms, while there is very little information on the influence of ILs on activated sewage sludge being a community of microorganisms [31,32]. Docherty et al. [4], in their studies of imidazolium and pyridinium-based ILs, used activated sewage sludge in small amounts only as an inoculum for standard OECD tests. Markiewicz et al. [33] also applied sewage sludge as inoculum for the modified OECD 301A procedure and, in addition, performed the determination of dehydrogenase activity (by TTC method) of sewage sludge. Moreover, they conducted sorption tests of ILs to the sludge flocs, which appeared to be an important process in neoteric solvent biodegradation as sorption was discovered to lower the ILs concentration in the solution. Another study on the influence of ILs on the dehydrogenase activity of sewage sludge activity performed by Liwarska-Bizukojc [34] revealed that 1-alkyl-3-methylimidazolium bromides with an alkyl chain of up to six carbon atoms do not inhibit dehydrogenase activity. In further research, Liwarska-Bizukojc et al. [35] showed that 1-alkyl-3-methylimidazolium bromides are poorly biodegradable, although their biodegradability increases with the elongation of the alkyl chain. Moreover, they observed that imidazolium-based ILs in the concentration of 50 mg·L^−1^ do not deteriorate the biodegradation of compounds present in municipal wastewater. However, it has to be pointed out that all of the described studies were performed on a small laboratory scale, while the growth dynamics and the system stabilization in the bioreactor can highly depend on the scale of the experiment.

As it was mentioned in our previous work [36], studies on bacteria adapted to tolerate different sources of xenobiotics were carried out; however, there are no or few reports of fungi in the biodegradation of ILs. Knowledge about the diversity of microorganisms, especially fungi found in sewage sludge, is incomplete and fragmented. It is estimated that currently, only 5% of fungi and 12% of bacterial species occurring in the natural environment are known. Genetic diversity of microorganisms in natural, extreme environments and various types of waste can be a source of strains with various biotechnological properties, used in new purification technologies or in improving the existing ones. In this regard, the conducted research aimed to extend the existing knowledge on the biodegradation of new-generation solvents by isolating the role of fungi in this process and the possibility of increasing their role in it.

It is extremely important to examine the possibilities of discharging and neutralizing ionic liquids, possibly with the use of fungal species, that can break down even exceptionally persistent xenobiotics; especially in the context of the increasingly discussed possibilities of using ionic liquids in industry or even as active pharmaceutical ingredients (APIs), and presented research aims to initiate this process.

## 2. Results and Discussion

### 2.1. Isolation and Identification of Fungal Species from Sewage Sludge

Twenty-two fungal isolates were obtained from the activated sludge. Pure cultures of fourteen isolates were obtained and identified (Table 1).

The phylogenetic tree of identified fungal isolates is presented in Figure 1. The tree structures revealed that the identified species were grouped in a few clusters of quite a close resemblance. All isolates identified had sequence identities of above 95% in comparison to the GenBank ones.

### 2.2. Minimal Inhibitory Concentration

As a result of the conducted measurements of the MIC of a particular IL, the influence of the alkyl chain length on the antifungal properties of the ionic liquid was found. ILs with a longer chain showed higher antifungal properties than those with a shorter chain. In the case of the liquid with the [Tf_2_N] anion, the limitation in estimating the MIC was the solubility of the compound in an aqueous solution. Therefore, the further part of the experiments focused on [OMIM][Cl], for which the highest antifungal propensity was recorded. The results of conducted MICs evaluation are presented in Table 2. The results correspond to those obtained by Łuczak et al. [39].

### 2.3. Biodegradation Tests

The study, conducted according to OECD guidelines, in duplicate systems with the 2 mM [OMIM][Cl] as tested substance and consortium of microorganisms as inoculum enriched with fungal strain, resulting in the average biodegradation equal to 64.93% within a 28-day testing period. For the individual fungal (*Candida tropicalis*) strain, the maximum of only 4.89% biodegradation rate was reached in 10 days, then inhibited. The results of the conducted biodegradability tests revealed that isolated individual fungal microorganisms are not able to metabolize 1-methyl-3-octylimidazolium chloride at a similar level as those combined into the community (e.g., with activated sewage sludge). Moreover, the community of *Candida tropicalis* with activated sewage sludge performed better in the biodegradability test than the single sludge alone. The influence of previously conducted fungus adaptation to the presence of ionic liquid also slightly increased the degree of xenobiotic degradation, especially in the initial stage. The results of the biodegradation test of the single fungal strain *Candida tropicalis* chosen on the basis of MIC results, as well as the community of microorganisms in the form of activated sewage sludge and fungal strain adapted or not adapted to the presence of [OMIM][Cl], are presented in Figure 2.

## 3. Materials and Methods

### 3.1. Chemicals

Neoteric solvents under investigation: [OMIM][Cl], [BMIM][Cl], [OMIM][Tf_2_N], and [BMIM][Tf_2_N] were purchased from IoLiTec ILs Technologies GmbH (Heilbronn, Germany); all of them were of 99% purity. ILs were degassed and dried under vacuum (20 Pa/24 h/70 °C), then stored in a desiccator. The description of the involved chemicals is presented in Table 3. Other chemicals involved are the following: buffer A: 60 mM NaHCO_3_, 250 mM KCl_2_, 50 mM TRIS, pH 9.5; buffer B: 2% BSA; Sabouraud Agar (BTL); PCR Mix Plus High GC (A&A Biotechnology, Gdańsk, Poland), PCR reaction primers proposed by White et al. [40] (Sigma, Poznań, Poland), ethidium chloride (Polgen, Łódź, Poland); Clean-up Concentrator (A&A Biotechnology, Gdańsk, Poland).

### 3.2. Isolation of Fungal Strains

Sewage sludge for experiments was collected from the aeration chamber of the “Wschód” municipal wastewater treatment plant in Gdańsk, Poland. The isolation of fungi was carried out directly from the activated sludge obtained from the “Wschód” WWTP, after which they were adapted to the presence of [OMIM][Cl] in order to improve their degradation ability. The direct inoculation of Sabouraud, not containing ILs with the initial samples, and incubation at two temperatures: 25 and 37 °C, for a period of two to 10 days was performed. Subsequently, media were inoculated by serially diluted initial samples and incubated as described above. In parallel, a five-day adaptation of microorganisms present in the initial sewage sludge samples to the presence of ILs in the amount of 25% minimal inhibitory concentration (MIC) against fungi was performed by incubating them at two temperatures: 25 and 37 °C for a period of two to 10 days. After incubation, fungal colonies were collected, and pure cultures were obtained.

### 3.3. Molecular Identification of Isolated Fungal Species

Isolation of fungal DNA was performed according to the Brillowska–Dąbrowska method [41]. Shortly, a small piece of mycelium was collected and placed in 1.5 mL Eppendorf tubes. 100 µL of Buffer A was added to each sample and incubated for 10 min at 95 °C. Subsequently, 100 µL of buffer B was added to each sample and immediately vortexed. Polymerase Chain Reaction (PCR) was performed with standard primers [40]: forward primer: ITS1 5′-TCCGTAGGTGAACCTGCGG-3′; reverse starter: ITS4 5′-TCCTCCGCTTATTGATATGC-3′). with the following temperature profile: pre-denaturation of DNA at 94 °C for 3 min; 35 cycles of 94 °C for 45 s; 55 °C for 45 s; 72 °C for 45 s). This was followed by a final primer extension at 72 °C for 10 min. The PCR products were sequenced (Genomed, Poland). The results were aligned against the NCBI database, and the species were identified (Table 1).

### 3.4. Minimal Inhibitory Concentration (MIC) Examination

Examination of MICs of four tested ILs for identified fungal strains was determined. 100 μL of ILs solution serially diluted ([OMIM][Cl]: 250–0.49 mM; [BMIM][Cl]: 1000–1.95 mM; [OMIM][Tf_2_N]: 1.4–0.03 mM; [BMIM][Tf_2_N]: 8–0.02 mM) were placed in the wells of the microtiter plate together with an inoculum of cultures of each fungal strain (OD600 = 0.6) and then incubated in 25 °C for 24 h. Fungal growth was determined with a microplate reader (Victor^3^V, Perkin Elmer, Centre of Excellence ChemBioFarm, Gdansk, Poland) at λ = 531 nm. MIC was evaluated as an 80% growth inhibition of tested strains, compared to the positive control (culture without IL).

### 3.5. Biodegradation Tests

The *Candida tropicalis* strain (in the concentration of 1 × 10^8^ CFU/mL) in a community with activated sewage sludge and [OMIM][Cl] (at a concentration of 2 mM) was selected for the biodegradation studies. Readily biodegradability studies were conducted using the OECD 301 F “manometric respirometry” test [42]. The test consists of measuring the rate of degradation of the substance in the environment under standard conditions reduced by the measurement obtained with a blank sample without the addition of the test substance. The resulting carbon dioxide (CO_2_) is bound by the absorber (sodium hydroxide). Thus, a pressure drop in the gas space of the closed reaction vessel is proportional to the respiration of the inoculum.

## 4. Conclusions

This is the first time that fungi isolated directly from the previously adapted to ILs activated sewage were used as an inoculum in the biodegradability test, both as a consortium and as individual species. As a result of the performed tests, the following fungal species: *Aspergillus terreus, Aspergillus fumigatus, Paecilomyces variotii, Penicillium adametzioides, Penicillium crustosum, Penicillium italicum, Geotrichum candidum/Galactomyces geotrichum* complex, *Aspergillus tubingensis, Candida tropicalis, Trichosporon domesticum, Candida glabrata, Trichoderma longibrachiatum, Candida sake*, and *Magnusiomyces capitatus* were selected from the activated sludge, showing particular resistance to the action of ILs, and having the ability to adapt to the xenobiotics tested.

A study on the mechanism of toxicity of ionic liquids to fungi was not the objective of the research presented in this publication. However, based on literature reports, proposed mechanisms of the effect of ionic liquids on fungi are available [43,44,45]. Potential mechanisms of toxicity of chemical compounds generally consist in affecting the cell membrane, changing its permeability and ability to transport proteins, inhibiting enzymatic activity and damage of genetic material [46,47,48,49]. Although the structure of ionic liquids classifies them as surfactants, it has been observed that their mechanism of toxicity to fungi may be more complex at the molecular level [50,51]. In literature reports, as a result of the conducted research, it was found that ionic liquids with short alkyl chains in the cation show a specific mechanism of toxicity by interfering with the metabolic pathways of fungi [41,45,52,53,54]. Comparative studies conducted on fungal metabolites have shown that sublethal concentrations of ionic liquids change the biochemistry of the cell [43,55]. The significant response observed was distinct from that induced by the presence of inorganic salts, ruling out the effect of oxidative stress as the primary cause of toxicity. The obtained response cannot be the result of co-metabolism in the case of imidazolium ionic liquids because no biodegradation of the imidazolium ring was observed in the literature under the conditions applied [8,56,57].

So, what is the role of fungi in the biodegradation of ionic liquids and is it crucial? In the presented research, the role of fungi as a single strain in the biodegradation of ILs has not been confirmed. On the other hand, as a result of bioaugmentation of the activated sludge with a selected fungus strain (*Candida tropicalis*), an increased degree of % degradation of the selected ionic liquid was obtained. Therefore, may the role of fungi be crucial in the IL’s biodegradation? Certainly, yes, due to the fact that they are an important element of the consortium of microorganisms (activated sewage sludge), which is able to cope with difficult-to-degrade compounds, and as was presented in this research, the consortium of organisms was the key factor in the proper degradation of the tested substance. On the other hand, the report of the research group of Petkovic et al. [45] reveals results of tests conducted on filamentous fungi that, adapted to the presence of the discussed xenobiotics, were able, as an individual species, to tolerate ILs concentrations in several orders of magnitude higher than those presented in this article (even 50 mM or higher). Although the authors of the article did not conduct biodegradation but only toxicity tests, they observed changes in the genetic structure of the tested strains, which suggests that fungi can adapt to the presence of ionic liquids, so perhaps with time and starting from lower concentrations it will also be possible to obtain a significant degradation of ionic liquids. Jets is certainly a topic worth further research.

In future research, it will also be extremely important to find the optimal conditions for the biodegradation of ionic liquids. Various culture conditions should be considered, such as a suitable inoculum consisting of organisms previously adapted to the xenobiotic or consisting of mixed cultures; and, in addition, appropriate media, pH, light, etc. It is also worth acquiring microorganisms living in extreme environmental conditions to check the possibility of their use in the biodegradation of a new generation of solvents, i.e., neoteric solvents, using their adaptation at the molecular level to the presence of usually toxic agents.

## Figures and Tables

**Figure 1 molecules-28-01268-f001:**
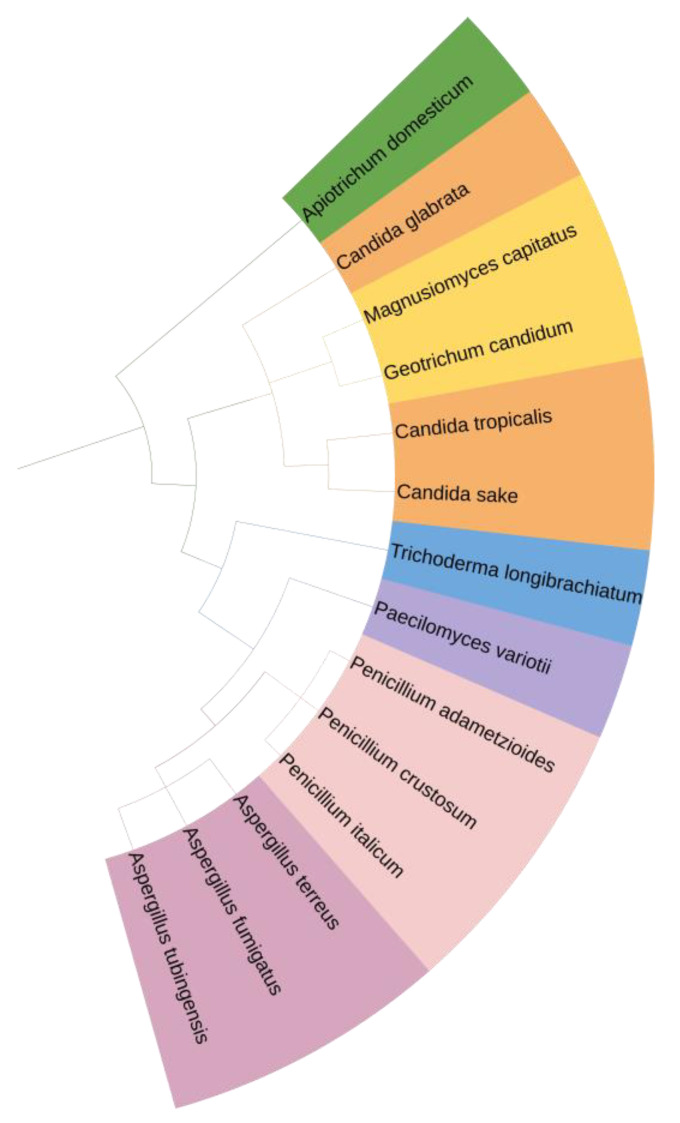
Phylogenetic tree of identified fungal species (visualization by iTOL reprinted/adapted with permission from Letunic and Bork [38], 2021, Oxford University Press).

**Figure 2 molecules-28-01268-f002:**
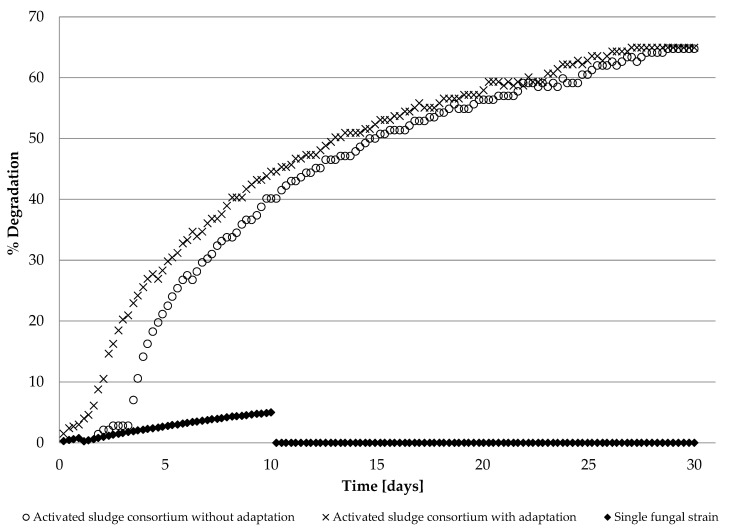
% Degradation test of [OMIM][Cl] by a single fungal strain *Candida tropicalis* (diamonds) and community of microorganisms in the form of activated sludge and fungal strain before (circles) and after (crosses) adaptation to the presence of IL.

**Table 1 molecules-28-01268-t001:** Fungal isolates from the activated sludge were identified by sequence comparison with GenBank [37].

Number	Fungal Strain
1.	*Aspergillus terreus*
2.	*Aspergillus fumigatus*
3.	*Paecilomyces variotii*
4.	*Penicillium adametzioides*
5.	*Penicillium crustosum*
6.	*Penicillium italicum*
7.	*Geotrichum candidum/* *Galactomyces geotrichum complex*
8.	*Aspergillus tubingensis*
9.	*Candida tropicalis*
10.	*Trichosporon domesticum*
11.	*Candida glabrata*
12.	*Trichoderma longibrachiatum*
13.	*Candida sake*
14.	*Magnusiomyces capitatus*

**Table 2 molecules-28-01268-t002:** Antifungal activity of neoteric solvents investigated against isolated fungal strains presented as MICs.

Fungal Strain	Antifungal Activity of Neoteric Solvents under Investigation Expressed as MIC [mM]			
	[BMIM][Cl]	[OMIM][Cl]	[BMIM][Tf_2_N]	[OMIM][Tf_2_N]
*Aspergillus terreus*	500	3.91	>1.4	>8
*Aspergillus fumigatus*	500	3.91	>1.4	>8
*Paecilomyces variotii*	15.63	<0.49	1	>8
*Penicillum adametzioides*	250	0.98	>1.4	>8
*Penicillum crustosum*	500	1.95	>1.4	>8
*Penicillum italicum*	125	0.49	0.7	>8
*Geotrichum candidum/* *Galactomyces geotrichum complex*	125	0.98	>1.4	>8
*Aspergillus tubingensis*	500	3.91	>1.4	>8
*Candida tropicalis*	15.63	31.25	1	>8
*Trichosporon domesticum*	500	3.91	>1.4	>8
*Candida glabrata*	31.25	0.49	0.7	>8
*Trichoderma longibrachiatum*	250	1.95	>1.4	>8
*Candida sake*	31.25	<0.49	0.7	>8

**Table 3 molecules-28-01268-t003:** Description of investigated neoteric solvents.

Abbreviation	Name	Empirical formula	Structure	Molecular Mass (g/mol)
[OMIM][Cl]	1-methyl-3-octylimidazolium chloride	C_12_H_23_ClN_2_	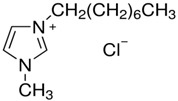	230.78
[BMIM][Cl]	1-butyl-3-methylimidazolium chloride	C_8_H_15_ClN_2_	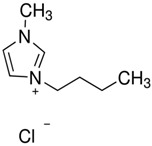	174.67
[OMIM][Tf_2_N]	1-octyl-3-methylimidazolium bis(trifluoromethylsulfonyl)imide	C_14_H_23_F_6_N_3_O_4_S_2_	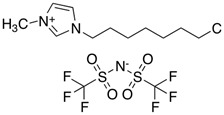	475.47
[BMIM][TF_2_N]	1-butyl-3-methylimidazolium bis(trifluoromethylsulfonyl)imide	C_10_H_15_F_6_N_3_O_4_S_2_	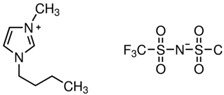	419.36

## Data Availability

The data presented in this study are available on request from the corresponding author. The data are not publicly available due to privacy reasons.

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
