# Peer review of "Role of Fungi in Biodegradation of Imidazolium Ionic Liquids by Activated Sewage Sludge"

_molecules, 2023, doi:10.3390/molecules28031268_

Round 1

Reviewer 1 Report

The manuscript entitled “Role of Fungi in Biodegradation of Imidazolium Ionic Liquids

by Activated Sewage Sludge” determines the role of fungi in biodegradation of ionic

liquids (ILs) as neoteric solvents. The work is interesting but cannot be recommended for publication in the present form until following points are addressed:

1.  1. The abstract is poorly written. It looks like the staring part of the introduction. Please rewrite the abstract including the following points: important of the topic, current challenges, objectives, brief methodology, main finding and concluding remarks.

2.      2. Introduction:

i.                    First two paragraphs are well known. Authors should summarize those within a few statements.

ii.                  The originality of the work is not highlighted properly.

iii.                There are too many lump references. Authors must describe the finding of such references.

iv.                Justify the selection of such two types of cations and anions. In fact, biodegradation of  selected first generation  ILs have  been well studied. It would be better to include some 2nd and 3rd generation ILs in this study.

3.     3.  Results & discussion

i.                    This section needs to be revised to improve the quality.  It seems that authors have just dumped some results.  In the most cases, no discussion is provided. The results must be supported by findings and literature data.

ii.                  Authors need to follow the common   practice in writing the   cation and anion.

For example, OMIM Cl is not in correct form.

iii.           The title mentioned “ the role of fungi on ILs biodegradation by Activated Sewage Sludge”. However,  this reviewer do not find any relevant  discussion.

4.      4. Methodology

i.                    Authors must provide appropriate refences when the methodology is adapted from other sources.

ii.                  Table 3, under column empirical formula authors mention S2. What does it mean?

5.       5. The conclusion is too long.  Conclusion needs to be rewritten highlighting the main findings related to the objectives of this manuscript.

Reviewer 2 Report

In their manuscript “Role of Fungi in Biodegradation of Imidazolium Ionic Liquids by Activated Sewage Sludge”, the authors isolated and identified of 19 fungal strains which is potentially capable of ILs degradation. They adjusted of the ILs concentration for biodegradability test and choosing strains with the highest biological robustness. They performed adaptation tests with both individual strains and community of microorganisms in order to acclimate the fungi to the presence of ILs in small concentrations and finally primary biodegradation tests. The manuscript provides new strains selection for the biodegradation of imidazolium ionic liquid. Based on the community of microorganisms, the mechanism of the strains synergy can be in-depth study. It will be more significance. Some questions: 1.About molecular identification of isolated fungal species, how did you choose your primers?  2. Sequencing alignment results are not present in support material.

Author Response

1.About molecular identification of isolated fungal species, how did you choose your primers? 

Ad.1. The primers have been chosen on the basis of Professor Anna Brillowska-Dąbrowska knowledge and experience as a standard primers. These are standard primers proposed by: 

White, T.J.; Bruns, T.D.; Lee, S.B., Taylor, J.W. Amplification and Direct Sequencing of Fungal Ribosomal RNA Genes for Phylogenetics. In PCR Protocols: A Guide to Methods and Applications, Innis M.A., Gelfand D.H., Sninsky J.J.; White T.J., Eds., Academic Press, Inc.: New York, U.S.A., 1990, pp. 315-322. http://dx.doi.org/10.1016/B978-0-12-372180-8.50042-1 The citation has been added to the proper section in the manuscript.

2. Sequencing alignment results are not present in support material.

Ad.2. That is correct. We have not submitted the alignment result in supported material. The alignment results will be added as a appendix.

Round 2

Reviewer 1 Report

Authors address all the questions arisen by this reviewer.